# OpenReview forum: "Uncertainty Drives Social Bias Changes in Quantized Large Language Models"
_ICLR.cc/2026/Conference — Submitted to ICLR 2026_

### Official Review · Reviewer_ghdj · 2025-10-17

**Soundness:** 3
**Presentation:** 2
**Contribution:** 2
**Rating:** 4
**Confidence:** 4

**Summary:**

This paper investigates how post-training quantization affects social bias in large language models through a large-scale empirical study. The authors curate QuantizedBiasBench by aggregating existing bias datasets and evaluate instruction-tuned models with 5 different quantization methods, resulting in various quantized variants. The core finding is that quantization induces what the authors term "behavior flipping," where up to 38% of responses switch between biased and unbiased states post-quantization. These changes are strongly correlated with model uncertainty, with high-uncertainty predictions more likely to flip than confident ones. The paper also reveals asymmetric impacts across demographic groups, where aggregate metrics mask substantial group-specific changes. The authors argue that 8-bit quantization is more stable than 4-bit methods and that model size provides no consistent robustness advantage.

**Strengths:**

+ The paper makes several valuable empirical contributions to understanding quantization's effects on model behavior. The scale of the evaluation is impressive, which provides a level of comprehensiveness missing from prior work. The identification of uncertainty as the primary driver of bias changes is an important finding that could inform future quantization strategies. The use of geometric mean probability for closed-ended response selection is a methodologically sound approach that addresses known issues with next-token probability selection and parsing failures.

+ The discovery of asymmetric social group impacts is significant from a fairness perspective. The paper demonstrates that aggregate metrics can mask substantial harm to specific demographic groups, which has serious implications for deployment. The finding that 8-bit quantization consistently shows 4-6× fewer behavioral changes than 4-bit methods provides practical guidance.

+ The methodology is rigorous, employing permutation-based significance tests with multiple comparison adjustments and effect size calculations. The authors' decision to ensure paired responses before and after quantization enables proper causal inference about quantization effects.

**Weaknesses:**

- The most significant weakness lies in what the authors claim as a primary contribution: QuantizedBiasBench. While the paper frames this as a novel unified benchmark, it is essentially an aggregation of 13 existing datasets without any new data collection, annotation, or curation beyond simple concatenation. Since quantized models use the same inference interface as their unquantized counterparts—only the underlying parameters differ—it remains unclear why existing benchmarks cannot be directly applied. The paper does not articulate what specific challenge necessitates creating this aggregate benchmark rather than simply evaluating on each constituent dataset independently. This raises questions about whether QuantizedBiasBench constitutes a genuine contribution or merely a repackaging of existing resources.

- The use of LLaMA Guard 3 8B for classifying bias in open-ended responses introduces a potential confound, as this classifier may itself be affected by the same biases being measured. The paper does not validate LLaMA Guard's reliability across different demographic groups or assess whether its judgments are consistent with human annotations. For datasets like FMT10K where the paper only evaluates the "Interference Misinformation" subset, the rationale for this selection is not adequately justified.

- The analysis conflates statistical significance with practical significance. While the paper reports that 11.4% of aggregate metric changes are statistically significant after multiple comparison correction, it does not establish what magnitude of bias change would constitute a practically meaningful harm. The focus on Cohen's d effect sizes helps but lacks grounding in real-world impact assessments. The paper also does not address how practitioners should interpret the finding that effects are zero-centered—does this mean quantization is "safe on average" despite high variance?

**Questions:**

Please see the weaknesses.

---

> ### Author Response · Authors · 2025-11-24
> **Response to Reviewer ghdj (1/2)**
>
> Thank you for your detailed review and recognition of our comprehensive empirical study, rigorous methodology, and important fairness findings. We address your concerns below.
>
> **Benchmark Framework Contribution**
>
> You correctly note that we aggregate existing datasets rather than collecting new data. However, our contribution is the **unified evaluation framework and standardized methodology**, not the raw data itself.
>
> Note that our framework enables more rigorous evaluation of changes in social bias due to LLM post-training modifications and can be used to quantify bias changes beyond quantization (e.g., alignment, supervised fine-tuning). We will rename QuantizedBiasBench to PostTrainingBiasBench to better reflect this, and clarify our contribution:
> 1. ***Standardized response extraction***: Prior studies use conflicting methods (next-token probability vs. parsing vs. full generation), leading to dropped responses and question asymmetries. Our geometric mean probability selection ensures all responses are extractable and comparable, eliminating a major source of bias in evaluation methodology.
> 2. ***Paired response evaluation***: Pairing responses pre- and post-quantization enables causal inference about individual response changes. Prior studies only compare aggregate statistics, missing the response-level flipping we document (Figure 3c: 21% flipping with stable aggregates). In the context of quantization, we are the first to evaluate both sub-group and individual-level changes.
> 3. ***Rigorous significance testing***: Permutation-based bootstrap with FDR correction across 546 dataset-axis-quantization combinations. Prior studies either lack significance tests or use inappropriate methods for paired comparisons.
> 4. ***Unified metric normalization***: All metrics rescaled to [0,1] with consistent directionality plus Cohen's d effect sizes, enabling direct cross-dataset comparisons impossible with heterogeneous original metrics.
> 5. ***Capabilities framework***: Conceptual organization by three fundamental capabilities (bias identification, equal outcomes, uncertainty preference) rather than dataset-specific artifacts.
>
> Why this matters: Prior quantization-bias studies (Table S1) use different datasets, extraction methods, and metrics, yielding conflicting conclusions. Our framework enables systematic evaluation revealing patterns (e.g., 8-bit stability, uncertainty-driven flipping) invisible in prior work. Existing benchmarks can be applied to quantized models, but **without methodological standardization, results are not comparable or interpretable**.
>
> We will revise Section 3 to emphasize: "PostTrainingBiasBench is not a new dataset but a unified evaluation framework with standardized methodologies enabling rigorous, comparable assessment of how post-training modifications affect bias."
>
> **LLaMA Guard 3 8B Validation**:
>
> You raise an important concern about classifier bias. LLaMA Guard 3 was validated for FMT10K, but not for BiasLens-GenWhy and CEB datasets. To address this, we conducted human validation on 400 responses (200 question pairs) stratified by bias shift occurrence, with results in Appendix A.6.
> 1. ***Paired evaluation improves reliability of predicting stability***: While individual response classification showed moderate performance (PPV = 86%, NPV = 70%), our paired evaluation framework improves reliability: NPV increases from 70% to 88% (up to 35pp gains for challenging datasets). High NPV (88%) confirms that when we report stable aggregates or responses, they are genuinely stable. We hypothesize that systematic biases (e.g., undercalling stereotyping) affect pre- and post-quantization identically and cancel in comparative analysis.
> 2. ***Paired evaluation trades off the ability to detect bias shifts for the ability to predict stability***. On the other hand, PPV drops from 86% to 64%, meaning we overcall changes in bias. While the PPV ranges from 0.85 to 0.9 for CEB-Conversation, overcalling bias shifts are likely true for BiasLens-GenWhy and CEB-Continuation for which the PPV ranges between 0.4 and 0.55. For low-prevalence phenomena like bias flipping, this creates substantial over-estimation: our reported flipping rates for BiasLens-GenWhy and CEB-Continuation are likely upper bounds, with true rates. Nevertheless, this does not impact the main findings of our paper, as we don’t make dataset-level claims about bias changes for BiasLens-GenWhy and CEB-Continuation in particular.
>
> Differential performance across demographic groups remains a limitation we acknowledge in our revised manuscript.

---

> > ### Author Response · Authors · 2025-11-24
> > **Response to Reviewer ghdj (2/2)**
> >
> > **Subset Justification**
> > 1. ***FMT10K Interference Misinformation subset***: FMT10K contains multiple task types (6 total). Full evaluation across all 6 tasks would require ~10K questions, making comprehensive evaluation infeasible given our scale (60 models × 85K total questions). The Interference Misinformation (IM) task was selected because it was the most challenging subset based on original paper results.
> > 2. ***BBQ subset selection***: Similarly, we evaluate only ambiguous-context BBQ questions (the more challenging subset where models exhibit greater bias) rather than all BBQ variants.
> >
> > We will add to Section 3.1 and Appendix A.4: "To maintain evaluation feasibility while maximizing diagnostic value, we select the most challenging and bias-relevant subsets from FMT10K and BBQ, following the original papers' recommendations for bias assessment."
> >
> > **Statistical vs. Practical Significance**
> >
> > You note Cohen's d helps but lacks real-world grounding. We agree that it is necessary to identify what amount of significant change is of practical value. As our results indicate, bias in real-world settings is context-dependent, affected by semantics and prompt structure. We believe it is beyond the scope of our study to definitively establish and merits its own study.
> >
> > A major contribution of our study is in the standardization of bias evaluation across benchmarks, which has been severely fragmented thus far. Cohen’s d effect sizes provide a means to quantify how strong a difference in aggregate metrics are relative to the amount of metric variation for a model on some dataset.
> >
> > Establishing what constitutes "meaningful harm" requires domain expertise, deployment context, and stakeholder input beyond our scope. We provide the statistical tools (effect sizes, paired testing) for practitioners to assess practical significance in their specific contexts.
> >
> > We believe this is a valuable point and will add to the limitations: "While we quantify effect sizes to enable comparison, determining practical significance thresholds requires domain-specific assessment considering deployment context, affected populations, and alternative options."
> >
> > **Interpreting Zero-Centered Effects**
> >
> > You ask whether zero-centered effects mean quantization is “safe on average”. We would like to clarify that this is an incorrect interpretation. Our findings suggest any quantized model instance effectively samples from a distribution of effect sizes, leading to positive or negative changes in aggregate measures. And even when aggregate metrics don't change, biases shift significantly at the subgroup level:
> >
> > 1. ***Hidden asymmetric harms***: Zero-centered aggregate changes can mask severe group-specific impacts. For example, aggregate change of ~0 can result from -14.1% improvement for one group and +18.6% deterioration for another (Figure 5b). The "average" user experiences no change, but specific demographic groups experience substantial harm.
> > 2. ***Deployment implications***: Practitioners cannot rely on coarse aggregate metrics to assess safety. They must: (1) examine group-disaggregated impacts, (2) assess high-uncertainty predictions separately, and (3) consider worst-case rather than average-case outcomes for relevant populations.
> > 3. ***Why bidirectionality matters***: The zero-centering explains conflicting findings in prior work (some studies report increased bias, others decreased). It's not that quantization has a net-neutral effect on bias - rather, effects are stochastic and heterogeneous across groups and questions, averaging to zero misleadingly.
> >
> > We will add this clarification in the conclusion.
> >
> > **Additional Improvements**
> > 1. ***Establishing causal effect of uncertainty changes***: While our benchmarking results showed a convincing relationship between uncertainty and response changes. We have further strengthened our causal claim with an interventional experiment (Section 4.5) that directly manipulates uncertainty to establish causality, addressing concerns raised by other reviewers.
> >
> > **Summary of Revisions**
> > 1. ***Rename to PostTrainingBiasBench*** and clarify contribution as evaluation framework, not dataset
> > 2. ***Validate LLaMA Guard*** through human annotation on stratified sample
> > 3. ***Add practical significance to limitations*** acknowledging context-dependency while providing tools for assessment
> >
> > These revisions preserve our core contributions: (1) unified framework enabling rigorous bias evaluation, (2) discovery of response-level flipping hidden beneath stable aggregates, (3) causal evidence linking uncertainty to flipping (Section 4.5), (4) documentation of asymmetric group impacts invalidating aggregate assessments, and (5) actionable deployment guidance (8-bit quantization, uncertainty screening).
> > Thank you for highlighting the need to clarify our methodological contributions and address the critical distinction between statistical and practical significance.

---

> > > ### Comment · Reviewer_ghdj · 2025-11-25
> > >
> > > Thanks for the response and clarification. I appreciate the authors' efforts to address the review feedback, and I acknowledge that I have fully read the author responses to this and other reviews. While I recognize the value of the proposed evaluation framework, my primary concern remains with PostTrainingBiasBench (previously QuantizedBiasBench) as one of the main contributions. Given that this benchmark is aggregated from multiple existing benchmarks without direct empirical evidence demonstrating clear advantages over evaluating these benchmarks separately, I will maintain my original score.

---

### Official Review · Reviewer_nrYG · 2025-10-19

**Soundness:** 4
**Presentation:** 3
**Contribution:** 2
**Rating:** 4
**Confidence:** 4

**Summary:**

This paper examines how model quantization affects social bias in large language models (LLMs). Rather than simply measuring performance degradation, the authors analyze how quantization can change bias direction and magnitude across demographic attributes. Using several bias benchmarks and multiple quantization levels, the study finds that quantization changes model bias patterns, sometimes amplifying, attenuating, or even reversing the direction of bias. The authors attribute these shifts to increased prediction uncertainty and representation distortion after quantization, arguing that uncertainty mediates how bias manifests. They further analyze correlations between uncertainty metrics and bias changes, and propose uncertainty-aware decoding as a potential stabilizer.

As the authors pointed out, the idea of connecting bias and quantization has been explored before in several papers. The main contribution to me is the strong execution of the empirical analysis and organized summary of how quantization reshapes biased behaviors.

**Strengths:**

Comprehensive empirical study and strong execution:
The paper conducts a systematic evaluation of how quantization affects bias behavior across multiple LLMs, quantization levels, and bias benchmarks. The experiments are well-organized and presented clearly, allowing readers to trace how quantization alters model outputs in nuanced ways. This level of empirical thoroughness is rare in the fairness–quantization intersection. The paper is well-written, transparent about its setup, and likely reproducible too.

Meaningful interpretation of the bias-quantization relationship and strong implications:
The paper’s empirical findings go beyond surface-level correlations: they reveal consistent, interpretable patterns in how quantization reshapes bias behavior. The observation that bias direction and magnitude shift systematically with changes in predictive uncertainty provides a concrete, data-driven lens for understanding compression side effects. These patterns can inform future research by serving as diagnostic signals and by motivating new approaches to quantization or calibration that explicitly preserve fairness properties. In this sense, the results themselves are a valuable contribution, mapping a space that future theoretical and mitigation studies can build upon.

Relevance and timeliness:
With growing interest in efficient LLM deployment, understanding fairness and bias behavior under quantization is practically important. This study provides a useful empirical foundation for the community, even if it lacks algorithmic innovation.

**Weaknesses:**

Mostly an empirical study and limited novelty:
The paper offers a careful empirical analysis but no substantial methodological or theoretical innovation. The link between quantization, calibration drift, and fairness degradation has been noted in prior work, and the “uncertainty drives bias change” framing mainly restates these existing insights.

No actionable mitigation or theoretical advancement:
The study refines measurement rather than introducing new modeling ideas or actionable mitigation strategies. This limits its scientific and practical contribution.

Scope limited to established benchmarks and settings:
The experiments rely on standard bias datasets and do not explore real-world, multilingual, or compositional contexts. As a result, the findings may not generalize to practical downstream applications where bias surfaces differently.

Correlation vs causation on the uncertainty causing bias change point: (which is actually one of the biggest contributions in this paper)
The paper’s central claim is that quantization -> higher uncertainty -> bias change. However, the experiments show that 1. quantization changes outputs, 2. quantization increases uncertainty, and 3. quantization changes bias. There doesn't seem to be a direct test showing causation. Maybe some type of controlled intervention, for example increasing uncertainty through noise or errors, and observing bias will prove this point more strongly.

**Questions:**

Causation point from above:
How can we distinguish this from a correlation where both uncertainty and bias shift due to shared representational distortions?
Have you attempted interventions that manipulate uncertainty directly (e.g., entropy regularization, confidence penalization) to observe causal effects on bias?

Robustness of findings:
To what extent can variance in bias metrics be statistically explained by changes in predictive uncertainty? Are the direction reversals stable across random seeds, datasets, and decoding strategies, or are they stochastic artifacts of quantization noise?
Do the observed bias shifts generalize across different quantization methods?
Can you provide qualitative examples where bias direction flipped, and explain what linguistic or representational changes may underlie the shift?

Higher level stuff:
The results suggest that quantization changes bias patterns, not consistently amplifies them. How should we interpret such reversals in terms of fairness?
Beyond post-hoc decoding, do your findings suggest principles for designing fairness-preserving quantization methods?

---

> ### Author Response · Authors · 2025-11-24
> **Response to Reviewer nrYG (1/2)**
>
> Thank you for your detailed review and recognition of our comprehensive empirical study and strong execution. We appreciate your thoughtful questions and address them below.
>
> **Establishing Causality Through Direct Intervention**
>
> You correctly identified the critical gap in our original submission: demonstrating that uncertainty → bias change requires controlled intervention. In response, we have conducted new experiments (Section 4.5; Figure 6) that directly manipulate uncertainty to establish causality:
> 1. ***Controlled intervention design***: We use preference optimization on Qwen 2.5 0.5B to causally manipulate pre-quantization uncertainty in two directions: SimPO reduces uncertainty by encouraging confident responses, while EntropyMax increases uncertainty by maximizing entropy between paired responses (5,322 BBQ questions).
> 2. ***Causal evidence***: Figure 6b-c demonstrates a clear dose-response relationship where manipulating uncertainty pre-quantization directly affects post-quantization response flipping. Increased relative uncertainty (entropy) and decreased absolute confidence (token probability) both causally increase response flipping rates, establishing that uncertainty drives flipping independent of shared representational distortions.
>
> This intervention addresses your concern about distinguishing correlation from causation: by directly manipulating uncertainty while holding other factors constant, we demonstrate that uncertainty is not merely a proxy for representational distortion but rather a causal mediator of quantization's effects on bias.
>
> We will revise Sections 1-4.4 to use correlational language ("strongly associated with," "predictive of") and explicitly frame Section 4.5 as providing causal evidence.
>
> **Mitigation Strategies**
>
> You noted the study lacks actionable mitigation. However, we argue our findings provide several practical deployment strategies:
> 1. ***Quantization method selection***: 8-bit quantization shows 4-6× fewer behavioral changes than 4-bit methods (Figure 4a), providing immediate guidance for practitioners balancing efficiency and reliability.
> 2. ***Necessity to benchmark post-quantization***. Our large-scale experiments demonstrate the need to measure subgroup performance after quantization. Aside from decreasing quantization strength, no particular quantization strategy or model size is safer. Altogether, these point at a necessity to understand quantization’s impacts as measured on examples most relevant to a user’s downstream use case.
>
> Furthermore, our study is the first to empirically quantify uncertainty in the context of measuring biases in LLMs and connect it to changes in model biases. This finding offers promise for future work:
> 1. ***Screening of low-certainty responses for potential bias in high-risk settings***.
> 2. ***Preserving uncertainty as a target in fairness-aware quantization***. In recent work by Lee et al. (2024), they use preference alignment (QDPO) to match the output token distribution of the quantized and full-precision model [1]. Future work can explore how similar strategies affects behavior flipping and the impact of asymmetric subgroup representation.
>
> We acknowledge our study is primarily diagnostic rather than prescriptive, but these actionable insights directly inform deployment decisions and motivate future quantization methods that explicitly optimize for preserving uncertainty.
>
> **Scope and Generalization**
>
> You note the scope of our benchmarks. We selected 13 established datasets that represent a diversity in capabilities measured and methods for bias elicitation (closed-ended, open-ended single/multi-turn). Our findings are likely general and suggest response flipping is likely to occur on any untested bias dataset albeit at different rates of flipping. Nonetheless, you are absolutely correct that the datasets selected may not capture all modes of bias. We will expand limitations to acknowledge: (1) findings are English-specific, (2) controlled benchmarks may not capture all real-world bias manifestations, and (3) intersectional contexts remain underexplored.

---

> > ### Author Response · Authors · 2025-11-24
> > **Response to Reviewer nrYG (2/2)**
> >
> > **Robustness of Findings**
> > 1. ***Variance of bias metrics explained by predictive uncertainty.*** Although we demonstrate causality of uncertainty changes on response flipping (Section 4.5), our results don’t suggest that uncertainty alone is enough to explain the variance of bias metrics, since a biased response can be made with either high or low certainty. Our claims on uncertainty are specific to how pre-quantization uncertainty impacts the chances of response changing post-quantization.
> >
> > **Bias Flipping**
> >
> > We thank the reviewer for their engaging and insightful questions related to our observations on bias flipping. We address them below:
> > 1. ***Stability across datasets and quantization methods***. Figure 4(a) shows how rates of behavior flipping vary greatly by dataset, and we hypothesize this has to do both with the semantics (e.g., subgroup-related bias) and how the prompt is structured. On BBQ, however, we do observe instances, where a model exhibits a tendency to change in bias towards a particular direction. Figure 5(b) shows how one model when quantized in any of the 5 methods can show a pattern of increasing bias for male (+7%), and from Figure 5(c), this directionality of bias change for the male subgroup appeared consistent across BBQ, BiasLens-GenWhy and FMT10K.
> > 2. ***Stability across random seeds and decoding strategies***. Since we select or generate responses deterministically, this eliminates concern for the stability of our empirical findings. Nevertheless, understanding bias changes in non-deterministic text generation is important, and this will require innovations in quantifying uncertainty that is temperature-independent and innovations in robust fine-grained detection of bias in generated text. We will add this to necessary future work.
> > 3. ***Interpretations***. Figure 2(b) suggested that quantization redistributes uncertainty rather than increases it or decreases it. Figure 5(b) established that subgroup-specific bias changes can be systematic with certain models displaying tendencies towards one direction. The dampening of these changes when averaged across models suggests that: 1) each model exhibits different vulnerability to bias changes for each social group that quantization can alter, and 2) no particular social group is vulnerable across all models and quantizations. These are findings based on BBQ dataset. Future work is required to definitively establish if certain social groups are more susceptible across all models. To accomplish this, datasets with more realistic variation in prompt structure would be necessary, whereas the datasets evaluated in this study are template-based and follow a similar format within-dataset.
> > 4. ***Qualitative examples***. Newly added Appendix A.12 provides examples of bias flipping in both directions, showing how quantization affects responses in unconstrained generated text—from introducing inappropriate associations (Example 2: HIV with transgender veterans), switching from disagreeing to supporting bias (Example 1) and vice versa (Examples 3-4).
> >
> > **Summary of Revisions**
> > 1. **Add Section 4.5** establishing causality through preference tuning intervention
> > 2. **Revise causal language** in Sections 1-4.4 to be correlational
> > 3. **Expand conclusion and future work** to highlight actionable mitigation strategies from findings and important future directions
> > 4. **Explicitly list limitations** in a limitations section
> > 5. **Add examples** of flipped responses in appendix
> > 6. **Add citations** on existing fairness-aware compression methods
> >
> > These revisions address your concerns while preserving our core contributions: (1) causal evidence that uncertainty drives response flipping, (2) comprehensive empirical documentation of quantization's asymmetric impacts at unprecedented scale, (3) actionable deployment guidance, and (4) design principles for fairness-preserving quantization.
> >
> > Thank you for the constructive feedback that strengthened our work substantially.
> >
> > **References**
> > [1] Lee et al. 2024. Improving Conversational Abilities of Quantized Large Language Models via Direct Preference Alignment. arXiv:2407.03051

---

### Official Review · Reviewer_7VRD · 2025-10-31

**Soundness:** 3
**Presentation:** 3
**Contribution:** 2
**Rating:** 6
**Confidence:** 4

**Summary:**

This paper investigates the effect of post-training quantization on the social bias of large language models. The authors selected a set of evaluation benchmarks and metrics and evaluated a large set of models on them. They show that model uncertainty is the primary driver of the "behavior flipping" phenomenon, where responses change post-quantization. The study demonstrates that aggregation-based metrics are not capable of showing the full extent of this behavioral change. The models do not simply become more or less fair; rather, quantization causes their behavior to change in more complex and asymmetric ways.

**Strengths:**

The main strength of this paper is its extensive study. The authors clearly invested significant time to investigate this phenomenon. They experimented with a large number of datasets, models, and quantization methods, and they evaluated the results using statistical tests. This thoroughness is valuable.

**Weaknesses:**

- The core idea, that model compression affects social bias due to uncertainty of the models, is not entirely new. Prior work, like Gonçalves & Strubell (2023), Zakizadeh et al. (2023) and Delobelle and Berendt (2022), has shown similar effects for distillation and other compression methods. This study feels like an extension of this known phenomenon to a wider set of quantization techniques. However, I appreciate that the authors clearly position their work among these related studies.

- The abstract's main message is difficult to understand quickly. The term "quantization-induced behavior flipping" is vague. I suggest the authors state their main finding more directly. For example, something like: "We find that aggregate bias metrics are misleading because quantization causes models to change their behavior completely, in ways aggregate scores do not capture."

- The main finding that increased model uncertainty leads to flipped decisions seems somewhat expected. When a model is compressed, it is logical that its least confident predictions would be the first to change. I am more curious about related factors, such as the effect of generation temperature. For example, are quantized models less stable, meaning more likely to change their answers in different sessions? Figure 2(b) seems to suggest the overall uncertainty distribution is similar, but this could be explored further.

- As a minor issue, I am not sure if it is intended, but in the references, the Qwen model technical report (cited as Qwen et al. in Line 253) lists the model name as the first author, which is an unusual format for a citation.


References:
- [Understanding the Effect of Model Compression on Social Bias in Large Language Models](https://aclanthology.org/2023.emnlp-main.161/)
- [DiFair: A Benchmark for Disentangled Assessment of Gender Knowledge and Bias](https://aclanthology.org/2023.findings-emnlp.127/)
- [FairDistillation: Mitigating Stereotyping in Language Models](https://arxiv.org/abs/2207.04546)

**Questions:**

- The paper states that high-uncertainty responses are more likely to flip. Based on this, I expected the "response changed" (blue) distribution in Figure 2(a) to be skewed to the right (high uncertainty) and the "response unchanged" (light orange) distribution to be skewed to the left (low uncertainty). However, the plots do not all show this clear separation. Could the authors clarify this? Also, please specify if the y-axis represents the raw count of samples or a normalized density (percentage).

---

> ### Author Response · Authors · 2025-11-24
> **Response to Reviewer 7VRD (1/2)**
>
> Thank you for your thorough review and recognition of our extensive study. We appreciate your feedback and address your concerns below.
>
> **Distinguishing study from prior distillation/compression work**:
>
> We acknowledge important prior work on bias in distillation and compression (Gonçalves & Strubell 2023; Zakizadeh et al. 2023; Delobelle & Berendt 2022). However, our contributions extend substantially beyond replicating these findings for quantization:
> 1. ***Uncertainty as a predictor of bias changes***. We are the first to systematically examine the relationship between model uncertainty and bias changes in compressed LLMs. Our geometric mean probability metric standardizes response selection across datasets while quantifying uncertainty, revealing that high-uncertainty responses are 3-11× more susceptible to quantization-induced changes than confident predictions—a predictive relationship absent from prior compression studies.
> 2. ***Response-level analysis reveals hidden instability***. Prior work focuses primarily on aggregate dataset-level metrics, with limited examination of subgroup- or response-level changes. We systematically document massive response flipping hidden beneath stable aggregates: up to 21% of responses flip classification despite no significant change in dataset-level metrics (Figure 3c). This failure mode—where aggregate stability masks individual-level instability—has not been characterized in the compression literature and has critical implications for deployment.
> 3. ***Cross-dataset patterns enable actionable guidance***. Prior work typically reports dataset-specific metric changes without identifying generalizable patterns. By standardizing measurements across 13 datasets, we identify consistent relationships: 8-bit quantization produces 4-6× fewer behavioral changes than 4-bit methods (Figure 4a), larger models show no robustness advantage, and social groups experience asymmetric impacts ranging from -14% to +18.6% within the same model. These findings provide concrete deployment guidance absent from previous studies.
>
> We will revise our related work and discussion sections to more clearly distinguish these contributions, emphasizing both our evaluation framework and the practical implications of hidden response-level instability.
>
>
> **Why this matters despite intuitive appeal**:
>
> We agree the uncertainty-flipping relationship seems logical and intuitive, yet this connection has not been established in the context of quantization. Our contribution lies in the rigorous quantification of this phenomenon. To further establish the causality of our claims, we add an experiment to show the causal impact of modulating uncertainty on response flipping (**new** Section 4.5). By intervening on model uncertainty via preference tuning, we demonstrate that uncertainty (both relative and absolute) displays a dose-response relationship, supporting our correlational experiments.
>
> **Temperature and generation stability**:
>
> Thank you for raising the point on generation temperature and cross-session stability.
> 1. ***Temperature Settings.*** For closed-ended selection, we use temperature = 1 (unscaled logits) to avoid artificially sharpening or flattening the probability distribution when computing geometric mean token probabilities. While temperature affects the sharpness of distributions, it does not alter token rankings and therefore cannot change which response is selected. Consequently, varying temperature would affect our uncertainty quantification but not our estimates of response flipping rates in closed-ended tasks. We acknowledge that improved uncertainty quantification methods that account for temperature effects remain an important direction for future work.
> 2. ***Cross-Session Stability.*** This is an excellent direction for future research. Our study employs deterministic generation (greedy decoding with temperature = 0), which precludes assessment of cross-session variability. We will add this to the limitations
> 3. ***Token-Level Distributions.*** We agree that analyzing top-k/top-p token distributions could provide valuable mechanistic insights. Recent work by Lee et al. (2024) examined quantization-induced token flipping in generated text, finding that token probabilities remain largely similar and that divergence occurs primarily at ambiguous tokens with small gaps between top-1 and top-2 probabilities [1]. This mirrors our findings at the sequence level for response selection. Understanding how and when token-level divergence propagates to bias-relevant behavioral changes remains an important open question. We will add this to future work

---

> > ### Author Response · Authors · 2025-11-24
> > **Response to Reviewer 7VRD (2/2)**
> >
> > **General Improvements/Clarifications**
> > 1. ***Improving clarity of main finding***. We will revise the abstract to state our main finding more directly:  "Aggregate bias metrics are fundamentally misleading for quantized large language models: quantization causes up to 21% of individual responses to flip between biased and unbiased states while aggregate scores remain unchanged, creating invisible harms that standard evaluations fail to detect." We will also update Figure 1 to more clearly enumerate our three key findings with visual emphasis on the discrepancy between aggregate stability and individual-level changes.
> > 2. ***Overlap in entropy values for flipping/non-flipping responses***. For Figure 2(a), you correctly noted the overlap between blue (response changed) and orange (response unchanged) distributions. This occurs because of dataset-specific patterns:
> >     - BBQ: The "unknown" response option often has near-zero probability initially, so there is often only probability mass between the stereotypical and anti-stereotypical responses. The normalized Shannon entropy when 2 of 3 options have maximum entropy is approximately 0.63 (which is where the distribution centers).
> >     - SocialStigmaQA: Models respond with near-certainty (entropy ≡ 0) to select "cannot answer," resulting in <1% flipping despite low uncertainty
> > 3. ***Clarifying the y-axis in Figure 2(a)***. The y-axis for Figure 2(a) shows normalized density (percentage of responses). We will update the caption to include this.
> > 4. ***Qwen citation format***. The Qwen 2.5 technical report lists Qwen as the first author followed by the team in the bibfile. We’ll fix this.
> >
> > **Summary of Revisions**
> > 1. Strengthen differentiation from prior compression work (Section 2.2)
> > 2. Revise abstract for clarity: directly state that aggregate metrics mask individual-level changes
> > 3. Update Figure 1 to more clearly enumerate findings
> > 4. Add to limitations/future work: cross-session stability, token-level distributions, stochastic generation
> > 5. Clarify Figure 2(a): add explanation of y-axis and dataset-specific patterns to caption and text
> > 6. Fix Qwen citation format
> >
> > These revisions will better convey why our findings matter beyond their intuitive appeal: we provide (1) quantified, actionable risk assessments, (2) causal validation of uncertainty driving reponse instability, and (3) practical deployment guidance.
> >
> > **References**
> > [1] Lee et al. 2024. Improving Conversational Abilities of Quantized Large Language Models via Direct Preference Alignment. arXiv:2407.03051

---

### Official Review · Reviewer_oEM7 · 2025-11-01

**Soundness:** 1
**Presentation:** 4
**Contribution:** 3
**Rating:** 4
**Confidence:** 2

**Summary:**

The authors investigate the role of post-training quantization methods in large language models (LLMs), and how quantization can affect the social bias of models, i.e. biases with respect to protected social groups. The authors propose a unified benchmark of 13 existing bias datasets, and evaluate 50 quantized models on this dataset across a broad range of post-quantization methods and representation lengths. The authors use this analysis to make several observations they claim to be novel:

1. The changes of social bias are asymmetric across different social groups
2. Up to 21% of responses flip between biased and unbiased states post-quantization, and this is invisible in aggregate metrics.
3. The amount of response flip between biased and unbiased states is correlated with model uncertainty and quantization strength
4. The amount of response flip is uncorrelated with model size.

**Strengths:**

* The paper is overall well-written and presented, with clear claims of contributions and explanation of methodology/experimental setting.
* The background is very well written and appears to cover much of the relevant literature on quantization and bias in LLMs (if not uncertainty/bias as discussed in weaknesses)
* The paper is very well motivated with post-quantization of LLM models being pervasive in application and research, although the study of bias in this context is far from a unique motivation at this point.
* The author's point that existing papers present conflicting findings is interesting and rings true, and motivates well a broader study of more quantization methods across more large models.
* Three "capabilities" or bias metrics were defined that appear reasonable, if not exhaustive.
* The main finding that the authors claim as novel, i.e. the correlation between uncertainty and bias, is interesting and intuitive.

**Weaknesses:**

* Several of the claimed contributions in terms of conclusions appear far from novel, in particular:
  - per-group evaluation differs from aggregate evaluation significantly --- this is what motivates all work fairness/bias analysis.
  - impact on different social groups is asymmetrical --- again this seems to be a consistent finding in other work

  I think if the authors instead claim to have replicated these findings at a larger/broader scale than existing work, that would be fair, but that's not how they frame it currently.
* The main finding that the authors claim as novel, i.e. the correlation between uncertainty and bias, might be intuitive but of course is very far from causal. And yet unfortunately much of the working in the paper is clearly claiming a causal relationship, e.g. on pg 5: "Our analysis reveals that uncertainty is the primary driver of quantization-induced bias changes". As clearly worded here, this is a massive and dangerous over-claim that is not supported by evidence, and is not the only time the wording suggests it.
* The proposed dataset is of course a meta-dataset, in that the authors have put together a dataset of existing datasets, and would not stand alone as a dataset contribution. I'm aware the authors do not seek this, and a meta-dataset is a valid contribution, but I believe it's important to be explicit as to the strength of the contribution on this front.
* There is existing literature linking confidence calibration/uncertainty and bias, for example "Uncertainty-based Fairness Measures" by Kuzucu et al. in 2023 suggests that a better metric of fairness is if a model has similar uncertainty across social groups. While I believe the author's observation is novel in the context of quantization, the authors have not made it clear enough in the paper why this is a significant finding given that we know uncertainty and bias are linked, and that quantization can change the uncertainty of a model.


Comments to improve the paper:
* Check all phrasing of claims of confidence/bias and insure they are rephrased such that there is no causal claim.
* Explain the less novel-sounding findings like aggregate metrics being different than per-group, in a context where they are potentially novel (e.g. scale of number of models/different quantization methodology, etc).
* I'm somewhat surprised that there is no effort to demonstrate that confidence calibration has an effect on bias as the authors themselves suggest for future work, even if it's in a smaller toy-like setting. This would go a long way further in making the observation and contribution significant.

**Questions:**

* What is preventing demonstrating the effect of confidence calibration in a toy-like setting? I'm simply trying to understand if there is a better way to explain why it wasn't in the scope of this work, as it's not clear to the reader as things stand why it wasn't done.
* Is there any evidence to support the existing wording of a causal link between uncertainty and bias?
* Why is this a significant finding in the context of quantization given the existing work pointing to the correlation between uncertainty and fairness/bias in the general case?

---

> ### Author Response · Authors · 2025-11-24
> **Response to Reviewer oEM7 (1/2)**
>
> Thank you for your thoughtful review. We appreciate your recognition of our presentation quality and interesting uncertainty-bias finding. Below we address your concerns.
>
> **Addressing Novelty Claims**
>
> We acknowledge per-group analysis is standard in fairness research. However, we are the first to do so in the context of quantization with the following unique contributions:
> 1. ***Scale***: Prior quantization-bias studies (Table S1) evaluated ≤3 models with ≤3 quantization methods on ≤6 datasets. We evaluate 50 model variants across 13 datasets (85K questions), revealing patterns invisible in smaller studies.
> 2. ***Response-level flipping masked by stable aggregates***: Our key finding (Figure 3c) is that up to 21% of responses flip between biased/unbiased states while aggregate metrics remain statistically unchanged. This highlights not only the necessity of subgroup analysis in post-quantization benchmarking but also the importance of understanding how post-training modifications fundamentally alter bias unpredictably at the response-level, and it represents a fundamental gap in prior evaluations.
> 3. ***Bidirectional asymmetry within identical architectures***: We show quantization creates changes ranging from -14.1% to +18.6% within the same model (Figure 5b).
>
> We will revise to clarify: "While per-group analysis is standard in fairness research, we are the first to document how quantization-induced response flipping remains hidden beneath stable aggregate metrics."
>
> **Establishing Causality**
>
> Thank you for pointing out that our earlier wording strongly implies causality between uncertainty and response changes. We will revise claims in Sections 4.1-4.4 to use wording such as "strongly associated with". As per your suggestion, we successfully designed an experiment to establish causality by intervening on model uncertainty (new Section 4.5; Figure 6):
> 1. ***Controlled intervention design***: We use preference optimization on Qwen 2.5 0.5B to causally manipulate pre-quantization uncertainty in two directions: SimPO reduces uncertainty by encouraging confident responses, while EntropyMax increases uncertainty by maximizing entropy between paired responses (5,322 BBQ questions).
> 2. ***Causal evidence***: Figure 6b-c demonstrates that uncertainty has a dose-response relationship with response flipping, where increased response entropy led to significant increases in response flipping. Similarly, we show that decreases in average token probability also led to increases in response flipping. This demonstrated two ways in which uncertainty (both relative and absolute) influence vulnerability to responses changes from quantization.
>
> **Benchmark Framework Contribution**
>
> You correctly note that we aggregate existing datasets in something of a meta-dataset. However, our contribution is the **unified evaluation framework and standardized methodology**, rather the raw data itself.
>
> More broadly, our framework enables more rigorous evaluation of changes in social bias due to LLM post-training modifications, and it can be used to quantify bias changes beyond quantization (e.g., alignment, supervised fine-tuning). Furthermore, the framework is designed to be easily adapted for future studies (e.g., space for error decomposition, new methods for evaluating bias in generative text). We will rename QuantizedBiasBench to PostTrainingBiasBench to better reflect this, and clarify our contribution:

---

> > ### Comment · Reviewer_oEM7 · 2025-11-28
> >
> > I want to thank the authors for their response and improvements. I will try to go through those points in the same order here:
> >
> > **Addressing Novelty Claims**
> >
> > I think the statement "While per-group analysis is standard in fairness research, we are the first to document how quantization-induced response flipping remains hidden beneath stable aggregate metrics." is still an over claim in itself.
> >
> > This describes all previous research showing that quantized models represent a different function on a per-sample basis rather than aggregate metrics, which is any research looking at quantization in the context of fairness/bias.
> >
> > I think if the authors incorporate the context of the scale of their experimentation that will make this claim appropriate however.
> >
> > **Establishing Causality**
> >
> > I'm a lot happier with the re-wording of the text around this, rather than claiming to have shown causality. While I give credit for trying to follow through on this with interventions, I believe it's fair to say your interventions are not replicating the scenario in which you would like to claim causality (in the initial quantization or quantized-aware training), but a potentially different context where you change the uncertainty after the fact, although I could be misunderstanding this. I recognize that going further would be extremely expensive however and not in the scope of your work.
> >
> > I think pointing out the correlation and potential causal factor is a significant contribution in itself though, I will be clear that my main issues with the original version of the paper was on the misclaim or poor wording that suggested a claim, rather than not showing causality.

---

> ### Author Response · Authors · 2025-11-24
> **Response to Reviewer oEM7 (2/2)**
>
> 1. ***Standardized response and uncertainty extraction***: Prior studies use conflicting methods (next-token probability vs. parsing vs. full generation), leading to dropped responses and question asymmetries. Our geometric mean probability selection ensures all responses are extractable and comparable, eliminating a major source of bias in evaluation methodology. Not only that, we use geometric mean probability relative to multiple options as a means for quantifying uncertainty; something not previously done in this literature.
> 2. ***Paired response evaluation***: Pairing responses pre- and post-modification enables causal inference about individual response changes. Prior studies only compare aggregate statistics, missing the response-level flipping we document (Figure 3c: 21% flipping with stable aggregates).
> 3. ***Rigorous significance testing***: Permutation-based bootstrap with FDR correction across 546 dataset-axis-quantization combinations. All prior studies lack significance tests and often report simply differences in absolute magnitude and ranking.
> 4. ***Unified metric normalization***: All metrics rescaled to [0,1] with consistent directionality plus Cohen's d effect sizes, enabling direct cross-dataset comparisons impossible with heterogeneous original metrics.
> 5. ***Capabilities framework***: Conceptual organization by three fundamental capabilities (bias identification, equal outcomes, uncertainty preference). This in addition to dataset-specific artifacts.
> 6. ***Subgroup evaluation***. While subgroup evaluation is not new in the broader fairness literature, in the context of quantization, we are the first to evaluate both subgroup- and individual-level changes. For datasets BiasLens-GenWhy and FMT10K, this required us to parse the social group from the prompts using GPT-4 (more details are in Appendix A.4.2).
>
> Why this matters: Prior quantization-bias studies (Table S1) use different datasets, extraction methods, and metrics, yielding conflicting conclusions. Our framework enables systematic evaluation revealing patterns (e.g., 8-bit stability, uncertainty-driven flipping) invisible in prior work. Existing benchmarks can be applied to quantized models, but **without methodological standardization, results are not comparable or interpretable**.
>
> We will revise Section 3 to emphasize: "PostTrainingBiasBench is not a new dataset but a unified evaluation framework with standardized methodologies enabling rigorous, comparable assessment of how post-training modifications affect bias."
>
> **Distinguishing Study From Existing Work in Uncertainty and Bias**
>
> Thank you for pointing out Kuzucu et al. (2023). While prior work has linked uncertainty to fairness using traditional metrics in classification tasks, our contribution is demonstrating this relationship in the quantization context with causal evidence (new Section 4.5) and actionable deployment guidance. We differentiate through: (1) evaluation across 13 diverse bias benchmarks rather than single fairness metrics, (2) controlled intervention establishing causality beyond correlation, and (3) practical findings for quantization deployment:
> 1. ***Quantization strength***: 4-bit exhibits 4-6× more behavioral changes than 8-bit, providing clear efficiency-reliability trade-off guidance.
> 2. ***Heterogeneous effects***: Impacts vary by question semantics and social group, demonstrating that downstream task-specific evaluation is necessary.
> 3. ***Hidden harms***: Response flipping occurs even with stable aggregates, invalidating standard evaluation practices.
> 4. ***Actionable uncertainty metrics***: Geometric mean probability enables practitioners to identify vulnerable predictions pre-deployment for targeted mitigation.
>
> Beyond quantization, our proposed paired evaluation framework can be easily applied and extended to measure bias changes from other model modification methods (fine-tuning, distillation).
>
>
> **Summary of Revisions**
> 1. Moderate causal language in Sections 1-4.4; foreground Section 4.5 as causal evidence
> 2. Clarify novelty around response-level flipping at scale and significance of current work
> 3. Add missing citations on uncertainty-fairness links
> 4. Better explain dataset as unified framework
>
> We hope these revisions better convey our core contributions: (1) response flipping is driven by uncertainty and can occur widely despite little changes in aggregate metrics, (2) a rigorous framework for understanding the impacts of post-training methods on social bias, and (3) practical guidance for safer quantization (e.g., using 8-bit quantization when possible).

---

> > ### Comment · Reviewer_oEM7 · 2025-11-28
> >
> > **Benchmark Framework Contribution**
> >
> > I would say that any paper with an empirical evaluation could claim to have proposed a framework for evaluation and standardized methodology. I think it's more accurate to claim to have proposed a meta-dataset as you did.
> >
> > **Distinguishing Study From Existing Work in Uncertainty and Bias**
> >
> > Adding the relationship to existing work in this area will strengthen this paper, as I believe the author's claims on contributions relative to the previous work are likely accurate except perhaps for the causality side.

---

### Author Response · Authors · 2025-11-24
**General Comment to All Reviewers and Area Chair (1/2)**

We thank all reviewers for their thoughtful and constructive feedback. Your comments have substantially strengthened our work. Below we summarize the major revisions addressing common themes across reviews:

**Section 4.5: Causal Intervention Study Added**

Multiple reviewers correctly identified that our claims about uncertainty driving response flipping required causal evidence beyond correlation. We conducted new experiments using preference optimization (SimPO/EntropyMax) to directly manipulate pre-quantization uncertainty on Qwen 2.5 0.5B (5,322 BBQ questions). Figure 6b-c demonstrates a clear dose-response relationship: manipulating both relative uncertainty (entropy) and absolute confidence (token probability) causally affects post-quantization response flipping rates. This establishes that uncertainty drives flipping independent of shared representational distortions. We have revised Sections 1-4.4 to use correlational language ("strongly associated with") and explicitly present Section 4.5 as providing causal evidence.

**LLaMA Guard 3 8B Validation Study Conducted**

To address concerns about classifier bias in open-ended evaluation, we manually annotated 400 responses (200 question pairs) across BiasLens-GenWhy, CEB-Continuation, and CEB-Conversation. While individual response classification showed moderate performance (PPV = 86%, NPV = 70%), our paired evaluation framework dramatically improves reliability: NPV increases from 70% to 88% (gains up to 35pp for challenging datasets). High NPV confirms that when we report stable aggregates or responses, they are genuinely stable. However, PPV drops to 64%, meaning we overcall changes in bias. For BiasLens-GenWhy and CEB-Continuation, PPV ranges between 0.4 and 0.55, creating substantial over-estimation of flipping rates for these datasets specifically. Nevertheless, this does not impact our main findings, as we don't make dataset-level claims about bias changes for these particular datasets. Critically, this improvement is evaluator-agnostic—systematic biases cancel by design when comparing paired judgments—providing a reusable methodology for future studies. Full results are in Appendix A.6.

**Clarified Significance of Framework: QuantizedBiasBench → PostTrainingBiasBench**

We clarify that our contribution is the unified evaluation framework and standardized methodology, not a new dataset. The framework enables rigorous assessment of bias changes from any post-training modification (quantization, fine-tuning, distillation). Our framework enables systematic evaluation revealing patterns (e.g., 8-bit stability, uncertainty-driven flipping) invisible in prior work. Existing benchmarks can be applied to quantized models, but without methodological standardization, results are not comparable or interpretable. Key contributions include: (1) standardized response extraction via geometric mean probability, (2) paired evaluation enabling causal inference about individual response changes, (3) rigorous permutation-based significance testing with FDR correction, (4) unified metric normalization with effect sizes, and (5) capabilities-based organization. Section 3 now emphasizes this contribution.

**Improved Clarity of Main Findings**

We revised the abstract to state our central finding more directly: "Aggregate bias metrics are fundamentally misleading for quantized large language models: quantization causes up to 21% of individual responses to flip between biased and unbiased states while aggregate scores remain unchanged, creating invisible harms that standard evaluations fail to detect." Figure 1 has been updated to more clearly enumerate our three key findings with visual emphasis on aggregate-individual discrepancies.

---

> ### Author Response · Authors · 2025-11-24
> **General Comment to All Reviewers and Area Chair (2/2)**
>
> **Distinguished From Prior Work**
>
> We better contextualize our contributions relative to prior work:
> 1. ***Prior compression work***. We are the first to systematically examine uncertainty as a predictor of bias changes, document response-level flipping hidden beneath stable aggregates at scale (50 model variants across 13 datasets, 85K questions), and identify consistent cross-dataset patterns: 8-bit quantization produces 4-6× fewer behavioral changes than 4-bit methods, larger models show no robustness advantage, and social groups experience asymmetric impacts ranging from -14% to +18.6% within the same model.
> 2. ***Uncertainty-fairness literature***: While prior work (Kuzucu et al. 2023) linked uncertainty to fairness in classification tasks, we demonstrate this relationship in the quantization context with causal evidence (Section 4.5) and actionable deployment guidance across 13 diverse bias benchmarks.
>
> **Additional Improvements**
> 1. ***Limitations Section***: Expanded to acknowledge scope (English-only, established benchmarks may not capture all real-world bias manifestations), differential classifier performance across demographics, and that determining practical significance thresholds requires domain-specific assessment considering deployment context
> 2. ***Future Work***: Cross-session stability, token-level distributions, stochastic generation, fairness-preserving quantization methods
> 3. ***Qualitative Examples***: New Appendix A.12 provides examples of bias flipping in both directions in generated text
> 4. **Figure 2(a) Clarification**: Added explanation of y-axis (normalized density/percentage of responses) and dataset-specific uncertainty patterns to caption
> 5. ***Interpretation of Zero-Centered Effects***: Clarified this does NOT mean "safe on average" but rather demonstrates that aggregate metrics systematically obscure asymmetric harms, where specific demographic groups experience substantial deterioration hidden by offsetting improvements for others
>
> **Summary of Core Contributions**
>
> Our revised manuscript better conveys three core contributions:
> 1. ***Causal evidence*** that uncertainty drives response flipping, which can occur widely (up to 21%) despite minimal changes in aggregate metrics
> 2. ***PostTrainingBiasBench***: A rigorous, evaluator-agnostic framework for assessing how post-training modifications affect social bias
> 3. ***Actionable deployment guidance***: 8-bit quantization, uncertainty-based screening, necessity of downstream task-specific evaluation
>
> We believe these revisions substantially address all reviewer concerns while maintaining the integrity and impact of our findings. Thank you again for your constructive feedback.

---

### Meta-Review · Area_Chair_d2p3 · 2025-12-25

**Summary:**

The paper establishes a benchmark PostTrainingBiasBench, which is aggregated from multiple current datasets, to evaluate the bias changes in quantized LLMs compared with their pre-quantized counterparts. The key findings include instance response flipping under stable aggregated metrics, which is correlated to model confidence and quantization strength. Even though this paper provides the first investigation on bias changes by quantization across various models and quantization techniques, major concerns towards the submission centre on the conceptual novelty, technical contribution, and incorrect claims about the causality between confidence and bias. Considering these, my recommendation is to reject.

**Reviewer Concerns:**

The major concerns raised by reviewers are as follows.

- oEM7: overclaimed contribution on findings and framework construction; incorrect claim of causality; lack of follow-up exploration on whether confidence calibration enhances fairness in quantized models.
- 7VRD: limited informativeness of findings (relationship between compression and uncertainty and social bias); vague wording.
- nrYG: limited conceptual novelty and theoretical advancement; limited scope; conflation of correlation and causality; robustness of findings.
- ghdj: overclaimed contribution of aggregated benchmark; lack of validation and justification on LLaMA Guard as bias classifier and incomplete experiment; unclear practical harm in real-world scenario.

The authors have accordingly revised the technical contribution of the benchmark, corrected the claims of causality to high correlation, and conducted additional experiments to explore the causal link between uncertainty and justify the use of LLaMA Guard as bias classifier.  However, a core concern of the limited contribution of the benchmark still remains, and the experiment design in section 4.5 does not reasonably support the causality claim (oEM7).

**Reviewer Scores:**

The reviewers would keep the original scores.

---

### Decision · Program_Chairs · 2026-01-26

Reject